# Influence of Asafoetida Extract on the Virulence of the Entomopathogenic Nematode *Steinernema carpocapsae* and Its Symbiotic Bacterium *Xenorhabdus nematophila* in the Host *Pyrrhocoris apterus*

**DOI:** 10.3390/microorganisms11071678

**Published:** 2023-06-28

**Authors:** Haq Abdul Shaik, Archana Mishra

**Affiliations:** 1Institute of Entomology, Biology Centre, CAS, Branišovská 31, 37005 České Budějovice, Czech Republic; 2South Bohemian Research Center of Aquaculture and Biodiversity of Hydrocenoses, Faculty of Fisheries and Protection of Waters, Institute of Aquaculture and Protection of Waters, University of South Bohemia in České Budějovice, Na Sádkách 1780, 37005 České Budějovice, Czech Republic

**Keywords:** asafoetida, *P. apterus*, *Steinernema carpocapsae*, *Xenorhabdus nematophila*, virulence, pro-phenol oxidase (PPO), phenol oxidase (PO), immunity

## Abstract

Nematode–microbe symbiosis plays a key role in determining pathogenesis against pests. The modulation of symbiotic bacteria may affect the virulence of entomopathogenic nematodes (EPNs) and the biological management of pests. We tested the influence of asafoetida (ASF) extract on the virulence of *Steinernema carpocapsae* and its symbiotic bacterium, *Xenorhabdus nematophila*, in *Pyrrhocoris apterus.* A total of 100 mg of ASF killed 30% of EPNs in 48 h, while *P. apterus* remained unaffected. The EPNs pre-treated with 100 mg of ASF influenced *P. apterus*’s mortality by 24–91.4% during a period of 24 to 72 h. The topical application of ASF acted as a deterrent to *S. carpocapsae*, lowering host invasion to 70% and delaying infectivity with 30% mortality for 168 h. Interestingly, *Steinernema*’s symbiotic bacterium, *Xenorhabdus*, remained unaffected by ASF. An in vitro turbidity test containing 100 mg of ASF in a medium increased the growth rate of *Xenorhabdus* compared to a control. A disc diffusion assay confirmed the non-susceptibility of *Xenorhabdus* to ASF compared to a positive control, streptomycin. Pro-phenol oxidase (PPO) and phenol oxidase (PO) upregulation showed that ASF influences immunity, while EPN/ASF showed a combined immunomodulatory effect in *P. apterus.* We report that ASF modulated the virulence of *S. carpocapsae* but not that of its symbiotic bacterium, *X. nematophila*, against *P. apterus*.

## 1. Introduction

Scientific research on biological pest control, which relies on bioagents and bioactive plant metabolites, aims to eliminate agricultural pests without harming other ecosystem elements. The use of entomopathogenic nematodes (EPNs), which parasitize a wide variety of insect genera belonging to the families Steinernematidae and Heterorhabditidae (Nematoda: Rhabditida), is promising [1]. A mutualistic relationship exists between EPNs and two different genera of bacterial symbionts, *Xenorhabdus* and *Photorhabdus*, which contribute to insect mortality [2,3] that occurs when EPNs’ infectious juveniles (IJs) enter a host through their natural openings and release lytic enzymes. These enzymes act as insect toxins and protect the IJs from the host’s defense system [4,5]. The host’s defense system initially fights against such pathogen invasions but eventually loses the battle, leading to death within three days of infection [6,7]. Moreover, by generating and releasing cadaver volatiles, EPNs boost the success of predation by attracting healthy herbivores [8].

EPNs release venom proteins with tissue-damaging and immunity-altering capabilities that are fatal to various insects [9]. EPNs are being employed as the natural biological agent of choice, with traits including habitat adaptation, broad insect host range, outstanding host search ability, mass manufacturing ease, and the ability to withstand some chemical pesticides. Particularly those classified as “bio-pesticides” can easily be made commercially available, offering a chance to cover the pest management gap left by the chemicals. It is known that biological agents act effectively as synthetic pesticides [10,11]. Unfortunately, the virulence factors of EPNs are less explored and investigated. This is due to the significant gaps in our knowledge of the metabolic mechanisms that may control their pathogenicity, i.e., host–nematode interactions, symbiotic bacterial virulence, host immunity, and environmental factors [12]. The interaction between EPNs/symbionts and any other external factors of interest may be additive, antagonistic, or synergistic. These interactions depend on the EPN species, insect host, application rate, and time of application. Studies focusing on such interactions that identify differences in the virulence factors of EPNs and their symbiotic bacteria both individually and together will improve this perspective.

Recent progress with bioactive metabolites, including asafoetida (ASF), a herbal source of chemicals with bioactive characteristics, has gained attention in pest control. This oleo-gum resin has been of human interest since it was discovered to have significant therapeutic and nutritional values. Recent studies have discovered a number of valuable assets, including antibacterial, anthelmintic, and insecticidal properties. ASF alone has been used as a bio-insecticide, e.g., oil extracts from *Ferula asafoetida* L. have been used as an insect pest repellent against thrips, feeding deterrent and as an insect oviposition barrier [13,14,15]. Co-examining the relationship between the herbal component asafoetida (ASF) and EPNs may be the most effective method for assessing the virulence and pathogenicity of EPNs against insect pests. This may be beneficial in line with the underlying essential parameters influencing the metabolic functionality of both biological agents. According to recent studies, administering insecticides to nematodes either reduced or impacted their virulence [16,17,18]. Most of the ASF/nematode co-interaction research has been conducted on plant or alimentary parasite nematodes, with no reports on EPNs [19,20]. Moreover, in field applications, farmers have been using ASF for centuries to increase crop output and protect plants from nematode infestations and root diseases. These investigations shed some light on the combined activities of these biological agents against agricultural pests and crops, although the scientific basis for their activity remains unclear [21].

This study is an attempt to elucidate the metabolic factors underlying the co-interaction of asafoetida with the entomopathogenic nematodes *Steinernema carpocapsae* and their symbiotic bacteria, *Xenorhabdus nematophila*, revealing their virulence in the host pest *Pyrrhocoris apterus.* We also focus on studying the modulation of immunity in *Pyrrhocoris apterus* during such co-interactions. These mechanistic linkages between ASF activity, EPNs, their symbiotic bacteria, and insect pests have not been explored earlier. By revealing the environmental co-factors influencing the efficacy of bioagents when combined, the results of this study may offer new conceptualizations for pest management approaches.

## 2. Materials and Methods

### 2.1. Animal Culture

*P. apterus* rearing: A stock culture of the firebug *P. apterus* (L.) (Heteroptera), resourced from wild populations collected at České Budějovice (Czech Republic; 49° N), was used for the present study. Larvae and adults of the reproductive (brachypterous) morphs were kept in 0.5 L glass jars in a mass culture (approximately 40 specimens per jar) and reared at a constant temperature of 26 ± 1 °C under long-day conditions (18:6 h light:dark). They were supplied with linden seeds and water. Freshly ecdysed adults were transferred to small 0.25 L glass jars (females and males separately) and kept under the same photoperiodic, food, and temperature regimes as those under which they had developed [22]. To have uniform materials, only 7-day-old males were used for the experiments.

*S. carpocapsae* culture: The entomopathogenic nematode *S. carpocapsae* was collected in a local field and then reared in laboratory conditions, according to [23]. *Galleria mellonella* (L.) (Lepidoptera, Pyralidae) caterpillars were used as hosts in the stock cultures. In either case, infective juveniles (IJs) were collected from the succumbed hosts and disintegrated in 1 mL of tap water. Nematodes were individually transferred to clean water and kept in a refrigerator as 1 mL aliquots for up to 7 days. All nematode manipulations were performed using a dissection microscope [24].

### 2.2. Asafoetida (ASF) Treatment, Dose Assessment, and Mortality

High-purity-grade asafoetida was purchased from SSP Super Fine Asafoetida Company, Banglore, Karnataka, India. The dried asafoetida powder (5 g) was soaked in MilliQ water (20 mL) overnight at room temperature, and the yielded suspension was used for all experiments. The concentrations and dosages of the extract were expressed as the crude amount of the dried oleo-gum resin used to prepare the stock solution [25].

The application of ASF to the nematodes was performed using the drop method. The procedure included the topical application of a suspension mix of 50 IJs in desired ASF concentrations of 50, 100, 200, and 400 mg (from stock) in a 100 μL water droplet, and the wriggling (surviving) IJs were counted after 24 h of incubation at room temperature. The infected juveniles were touched with a needle to confirm death. The dose and mortality parameters were further assessed. The control nematodes were kept in 100 μL of water.

Adult *P. apterus* were intraperitoneally (IP) injected with 2 μL of ASF (50, 100, 200, and 400 mg) at the thoracic segment. The insects were then left for 24 h at 27 °C. The mortality rate and nematode virulence were recorded at 24 h, 48 h, and 72 h. For combined ASF and nematode treatments, a control (water) and ASF-treated nematodes (100 mg) were topically applied to *P. apterus*, and mortality data were recorded at 24, 48, and 72 h. The controls were considered mean representatives during all treatments.

### 2.3. Nematode Deterrent Effect and Virulence on ASF-Treated P. apterus

The adult *P. apterus* were IP injected with ASF (2 µL; 100 mg) and monitored for 24 h. Around 10 insects were placed in a 10-diameter petri dish, and the wet filter paper was sprinkled with 100 nematodes pre-treated with 100 mg of ASF for 24 h. In this experiment, our main aim was to observe the deterrent responses of the nematodes toward ASF-containing insects. They were left for 7 days on the same dish. Then, the insects were dissected and checked for the number of nematodes that had entered the cadaver.

### 2.4. S. carpocapsae Growth, Infectivity, and Virulence Assay on Host P. apterus

*P. apterus* adults were treated with 2 µL of ASF (IP-injected) and left at room temperature for 24 h. The control insects were treated with 2 µL of water. As EPNs are not evenly distributed in nature, another 100 IJs were placed on the wet filter paper to establish contact with the experimental and control insects, considering their spatial distribution (Abd-Elgawad, 2021) [26]. The adult mortality was monitored for 10 days to determine the nematodes’ pathogenicity. The number of nematodes that entered the cadaver was counted and analyzed for infectivity (%), while nematode growth in the cadaver was measured by counting the worms under a microscope.

### 2.5. Isolation of Symbioitc Bacterium X. nematophila from P. apterus, Turbidity Test, and Its Virulance on P. apterus

*Xenorhabdus nematophila*, a symbiotic bacterium, was cultured in vitro by injecting 10 IJs into the abdomen of *P. apterus.* After 24 h, one leg was punctured, and the hemolymph was streaked on nutrient agar (NA) plates with 0.8% Difco Bacto nutrient broth and 1.2% agar [27]. The plates were incubated at 25 °C. Characteristic colonies of *X. nematophila* developed after 3 days and were subcultured 2–3 times until a homogenous culture was obtained. The culture was transferred to nutrient agar slants and incubated at 25 °C for 3 days, after which it was stored at 6 °C and subcultured at least once every month. The impact of ASF was then examined using these bacteria. Initially, the bacteria were treated with broth containing 100 mg of ASF for a period of 24 to 72 h. Every 18 to 24 h, spectrum metric readings at a wavelength of 600 were recorded, and a graph was prepared to assess the growth of *X. nematophila* obtained from *P. apterus*’s hemolymph pure culture broth. The broth was initially centrifuged and pelleted, and 500 µL of distilled water was added to the pellet. The pellet was then vortexed and used for the *P. apterus* injections (2 µL) and a virulence test. For the combined treatments, we used distilled water with *X. nematophila* mixed with ASF.

### 2.6. Disc Diffusion Assay: Asafoetida Antimicrobial Activity Assessment Using Xenorhabdus spp.

For the ASF antimicrobial tests, the disc diffusion method using *Xenorhabdus* spp. bacteria was employed. This entomotoxic organism is symbiotically associated with the nematode *S. carpocapsae*. The bacteria were isolated from *P. apterus* adults infected with *S. carpocapsae* (IJs) [28]. The deceased *P. apterus* were opened using sterile needles and scissors after being surface sterilized in 75% alcohol for 10 min. NBT agar plates were streaked with a drop of the leaking hemolymph, and the plates were incubated at 30 °C in the dark for 24 h. Subsequently, a single bacterial colony was selected and streaked onto a new plate of NBT agar, after which it was used for the inoculation of a 2% LB broth (Lennox) solution. The inoculated solution was agitated at 30 °C at 150 rpm for a day. The following day, 0.2 mL of the bacterial solution was swabbed onto the agar plates with a density adjustment of 0.8 McF (McFarland’s bacterial density). Sterilized paper discs (3) were placed on the symbiotic bacterial plate along with 10 µL of 100 mg of ASF and streptomycin (30 µg) (Sigma-Aldrich; St. Louis, MO, USA). They were dried using laminar airflow and placed on the bacterial lawns. Only 10µL of sterile water was used on the paper discs for the controls using separate plates. The growth inhibition zones around the paper discs were measured, and their areas were determined for both the control and experimental plates after an overnight incubation at 30 °C.

### 2.7. P. apterus Immunity Assessment: Pro-Phenol Oxidase (PPO) and Phenoloxidase (PO) Assays

*P. apterus* hemolymph collection: After washing the *P. apterus* adults with 70% ethanol for disinfection, the prolegs and antennae were cut with a needle, and the hemolymph that leaked was collected into a 1.5 mL microtube that was maintained on ice. The collected hemolymph was diluted 20 times with cold 10 mM PBS buffer, pH 7.0, and centrifuged for 15 min at 1200 g at 4 °C. Subsequently, 95 μL of the supernatant was transferred into a microplate well containing 5 μL of dH2O to measure the active PO or 5 μL of chymotrypsin solution (Sigma-Aldrich; St. Louis, MO; 5 mg/mL of dH2O) to measure the active PPO [29]. In both cases, 50 μL of L-Dopa solution (the substrate for the PO; Sigma-Aldrich; St. Louis, MO, USA; 3 mg/mL of dH2O) was added to the mixture, and the colorimetric reaction was followed in a microplate reader (spectra MAX 340pc; Molecular Devices, San Jose, CA, USA) at 30 °C. Readings were taken for 30 min at 490 nm. The enzyme activity was analyzed using the software SOFTMax^®^Pro 5.2 (Molecular Devices, San Jose, CA, USA) and measured as the slope (Vmax value) of the reaction curve during the linear phase. The concentration of proteins in the hemolymph was determined using the bicinchoninic acid method described by [30]. For the calibration curve, bovine serum albumin was used. The activities were expressed in units of transmission density of the incubation mixture during the reaction per 1 min and per 1 mg of protein.

### 2.8. Data Presentation and Statistical Analyses

The results were plotted, and the statistical analyses were calculated using the graphic software Prism (Graph Pad Software, version 6.01; San Diego, CA, USA). A comparison of the groups to find statistically significant differences was performed using a one-way ANOVA at a significance level of *p* < 0.05. Tests of the homogeneity of variances confirmed the normal distribution of the dataset. The results of F-tests included degrees of freedom and degrees of freedom of the error (within-group degrees of freedom), and, subsequently, the means were separated using Tukey’s multiple comparison test. The one-way ANOVA, Dunnett’s multiple comparisons tests, and *p* < 0.05 were performed for the disc diffusion antibiotic effect. Points in the bar graphs represent the means ± SD. The numbers of replicates (*n* = number of replicates) are depicted in the figure legend.

## 3. Results

### 3.1. Asafoetida (ASF) Dose and Toxicity Assessment with Nematodes

Nematode mortality was found to increase with increasing the ASF dose. A one-way ANOVA of the drop method application revealed a significant difference in a dose-specific pattern (*p* < 0.0001, F = 184.9). Nematode mortality was significantly increased with the 100, 200, and 400 mg treatments when compared to the control (*p* < 0.01; *p* < 0.0001), respectively. The minimum nematode mortality was recorded with the 100 mg treatment (30%), while the maximum (98%) mortality was recorded with the 400 mg treatment. No significant mortality was seen with the 50 mg ASF treatment (Figure 1) when compared to the control. The 100 mg ASF dose with the least significant nematode mortality rate (approx. 30%) was assessed for further experiments.

### 3.2. Asafoetida (ASF) Dose and Toxicity Assessment with Firebug P. apterus

*P. apterus*’s survival remained unaffected (*p* < 0.001; F = 2.1) by the different doses of ASF treatment. The control exhibited 5% mortality, and the highest mortality of 6–7% occurred with the 400 mg ASF treatment. All treatments were non-significantly different in comparison to the controls (Figure 2).

### 3.3. Combined Effect of ASF and Nematodes on Firebug P. apterus Mortality

The combined interaction of ASF with the nematodes and *P. apterus* remained highly significant with *P. apterus*’s mortality (*p* < 0.0001; F = 192) when compared to the control (water + nematodes without ASF). At 24 h, a 24% mortality rate was observed for *P. apterus* (*p* < 0.001), which was significantly increased after 48h (61.2%) and 72 h (91.4%), respectively (Figure 3). It is interesting to note that 72 h were needed for the cumulative effect of the ASF-treated nematodes on *P. apterus*’s mortality (%) to match the controls. This strategy suggests that the combined mortality effect was drastically reduced when compared to the controls, indicating that mortality was delayed in response to ASF.

### 3.4. Combined Effect of ASF Pre-Treated P. apterus on Nematode Infectivity and Deterrant Effect

The deterrent responses of the nematodes toward the ASF-pre-treated *P. apterus* were significant (*p* < 0.001). The infectivity of the nematodes was recorded to be 30% after 10 days of treatment (Figure 4A). On the other hand, the ASF-treated nematodes showed reduced virulence with extended *P. apterus* survival (Figure 4B) during a period of 7 days. Mortality (%) remained as follows: 3.25, 7, 11.75, 15.5, 19.74, 25.5, and 28.5 at 24, 48, 72, 96, 120, 144, and 196 h, respectively.

### 3.5. Effect of ASF on Symbiotic Bacteria X. nematophila Growth, Turbidity Test, and Its Effect on P. apterus Mortality

The growth of *X. nematophila* was significantly impacted by the ASF treatments (*p* < 0.0001). More than 50% turbidity with the control and only ASF was recorded with the media containing *X. nematophila*, which significantly increased to another 14% when ASF was added to the culture medium (Figure 5A; *p* < 0.001).

In vitro tests were performed by injecting *X. nematophila* into *P. apterus* (%), which showed similar results. Significantly increased mortality was observed for the *X. nematophila* and ASF+ *X. nematophila*-treated *P. apterus* when compared to the control and ASF treatment groups (Figure 5B; *p* < 0.001).

### 3.6. Growth of Symbiotic Bacteria X. nematophila on ASF: A Disc Diffusion Assay

The disc diffusion assay indicated a potential antibiotic effect on the *X. nematophila* bacteria. Significantly high inhibition zones around the disc with the positive control, streptomycin (*p* < 0.001), indicated bacterial growth inhibition, whereas ASF showed no inhibitory zone, which remained insignificant in comparison to the controls (Figure 6; *p* < 0.05).

### 3.7. Effect of ASF on Pro-PO and PO on P. apterus: Assessments of Immunity Markers

Pro-PO showed a pattern of significant increases with the treatment groups (*p* < 0.0001; F = 53.01). The highest absorbance was recorded with the nematode treatment, which significantly activated the enzyme (*p* < 0.001). IJs and combinations of IJs and ASF treatments showed a significant high titer when compared to the control group (*p* < 0.001) (Figure 7A).

The relative activity of the phenol oxidase enzyme (PO) significantly varied between the treatment groups (*p* < 0.0001; F = 159.7). A significant high absorbance in the hemolymph was recorded with the treatment groups of IJs (*p* < 0.001) and ASF (*p* < 0.001). During the ASF treatment, the PO activity remained similar to that of the control. The highest absorbance value was recorded for the combined application of IJs and ASF (Figure 7B).

## 4. Discussion

We investigated whether ASF had an impact on the virulence of EPNs and the bacteria that coexist with them. If so, how are pest immunity and mortality affected by the consequences of pest control?

We explored various elements underpinning ASF exposure and EPN virulence against the pest *P. apterus.* The main attentive factors were as follows: (a) EPNs’ host preference ability or by ASF influence; (b) EPNs/bacterial symbionts’ host killing potential or by ASF influence; and (c) host immunity factors by EPNs or ASF influence.

The ASF-co-exposed nematodes showed prolonged survival of *P. apterus*, indicating that the nematode was attenuated with ASF and recovered over time, causing delayed infectivity and mortality. Under environmental conditions, it is known that the nematodes undergo developmental arrest, where they become dormant quickly and survive longer until they find favorable conditions. Through a process of recovery [31], the nematode overcomes unfavorable conditions, in this case, ASF, and begins its mature phase in the host by feeding host nutrients to its symbiotic bacteria. It may usually increase the host’s respiration and, accordingly, mitochondrial respiratory chain deficiency, resulting in its developmental arrest, as documented in *C. elegans* [32,33].

In the topical application of the nematodes, where *P. apterus* was pre-treated with 100 mg of ASF, the nematodes’ entry into the host was reduced significantly. The plausible reason that we suggest is that the presence of ferulic acid and its esters, sulfur-containing compounds, monoterpenes, and other volatile terpenoids in ASF acted as a feeding deterrent, limiting the entry of the nematodes into the host [34,35]. According to a prior study, ASF also has nematicide properties, inducing nematode death at ASF concentrations of 0.5 mg/mL [13]. As a result, it can be assumed that the EPNs may have experienced similar consequences with limited host invasion. Another intriguing study strategy involves the emphasis placed by some researchers on the potential for a specific attenuation of chemosensory pathways for nematode control [36,37,38,39]. This motion also concurs with the study’s result.

When we evaluated the preference of the ASF-pre-treated *P. apterus* (100 mg) for host invasion, the findings were startling. Following 7 days of co-interaction, the exposure of the nematodes to *P. apterus* resulted in a dramatic reduction in the EPNs’ pathogenicity to a limited maximum of 35% infection. This indicates that co-factors underlying EPNs’ dormancy, mortality, and extended time of contact likely played a crucial role in controlling the nematodes’ virulence. As a result, our findings showed that the virulence factor was negatively impacted by the co-interaction of ASF-pre-treated bugs and EPNs, extending the lifespan of the pest *P. apterus*.

It would be crucial to determine whether the nematodes or their symbiotic bacteria, *X. nematophila*, were accountable for the decreased virulence of the ASF-co-exposed EPNs. The most astounding finding we found was that, in contrast to the controls, the growth and proliferation of *X. nematophila* were not adversely affected by the ASF treatments. This makes clear the understanding that only EPNs were adversely affected when co-exposed to ASF. Similar findings were observed in *Pseudomonas aeruginosa* and *B. subtilis*, which are non-susceptible to ASF powder [40]. Some contrasting studies reported that volatile oils from *F. asafoetida* promote antibacterial activity and that the rate of bacterial growth reduction is directly proportional to the concentration of ASF [41]. Mishra and Behal [42] suggested that alcoholic and water extracts of ASF exhibit antimicrobial activity in the cases of *B. subtilis*, *S. aureus*, *E. coli*, and *Penicillium aeruginosa*. With our data, we report that *X. nematophila* showed no antimicrobial activity within our utilized ASF concentrations. One such study by Moghadam et al. (2014) [43] reported the minimum inhibitory concentration (MIC) and minimum bactericidal concentration (MBC) of *F. asafoetida. E. coli* showed an MIC value of 1562.5 µg/mL and an MBC value of >100,000 µg/mL. We can assume that the ASF concentrations used in this study were effective against nematodes rather than *X. nematophila*, as they were not able to inhibit the growth of the latter. ASF-treated *X. nematophila* showed no change in virulence when compared to the controls. A similar result by Charu Singh et al. [40], where ethanol and acetone extracts of ASF showed no antibacterial effect on *Pseudomonas aeruginosa*, supports our observation.

In addition, co-exposed ASF/EPNs showed upregulation of their innate immunity markers PPO and PO, indicating their essential roles in defense and insect immunity. One of the initial processes of an immune response to bacteria is the activation of the pro-phenol oxidase cascade. In a study with similar findings, PPO and PO were increased when *X. nematophila*, a symbiont of the nematode *S. carpocapsae*, was introduced into *Drosophila* [4]. With the help of our study’s data, we produced some novel and intriguing findings that will advance our understanding of nematode–pest pathogenicity. Moreover, we are far from a complete understanding of the physiological and biochemical mechanisms underlying the herbal bioactive metabolite ASF’s functionality in insect immunity. We propose a further in-depth molecular study on this untouched topic to dispel myths and facts about the use of a combination of bioactive agents (herbal bioactive compounds and EPNs) that may be advantageous for the environment and pest management.

## 5. Conclusions

Overall, the bioagent’s virulence against pest control is always influenced by environmental co-factors. It is now understood that EPNs with symbiont bacteria have an unusual defense mechanism when exposed to bioactive herbal metabolites. In this study, we report that the duo’s (EPNs/symbiont bacteria) pathogenicity is influenced by a number of factors, including ASF concentration, duration, contact conditions, and host. Our findings also conclude that nematodes undergo attenuated and slow recovery when co-exposed to ASF. The ASF dose and exposure tenure will directly relate to EPNs’ mortality rates. We tested the symbiotic bacterium’s response to ASF but found no inhibitory effects on the bacteria’s virulence, growth, or proliferation. We conclude that during ASF-derived bioactive compound interactions, EPNs and the bacteria that live in their symbiotic relationship function independently.

## Figures and Tables

**Figure 1 microorganisms-11-01678-f001:**
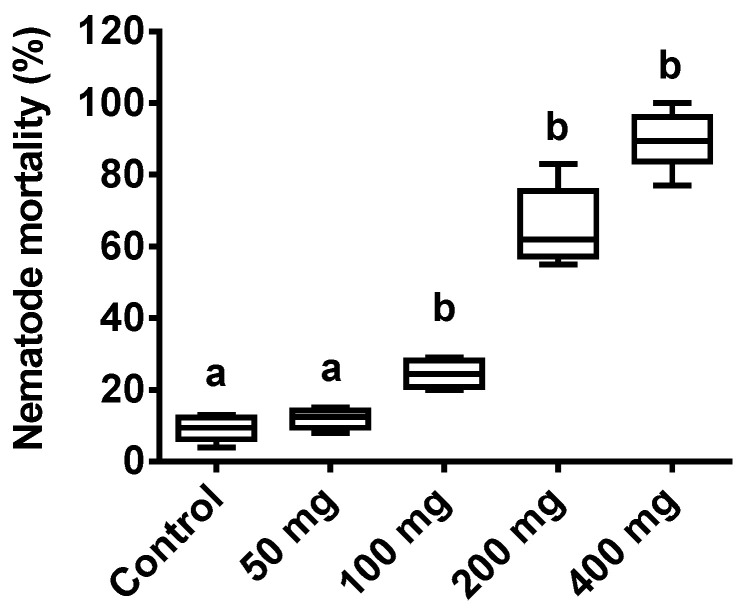
Effect of ASF treatments on nematodes. Percent mortality of nematodes with four different ASF doses and a control with water treatments using the drop method. Results are expressed as means ± standard deviation (SD). Data represent the means of three replications (*n* = 50). Values (represented as a, b) with different superscripts are significantly different (*p* < 0.05).

**Figure 2 microorganisms-11-01678-f002:**
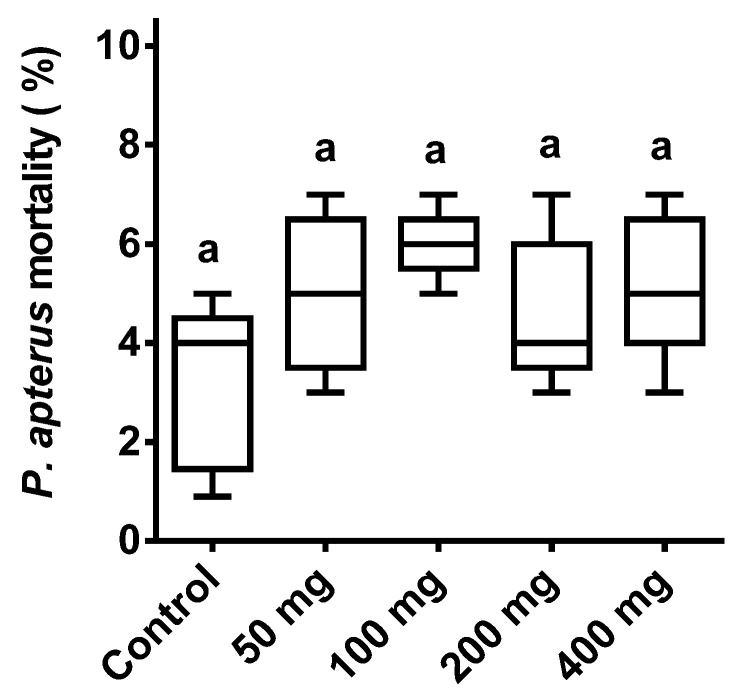
Effect of ASF treatment on *P. apterus*. Percent mortality of *P. apterus* with four different ASF doses and a control with water treatments using IP injection. Results are expressed as means ± standard deviation (SD). Data represent the means of three replications (*n* = 50). Values (represented as a, b) with the same superscripts remained non-significantly different (*p* < 0.05).

**Figure 3 microorganisms-11-01678-f003:**
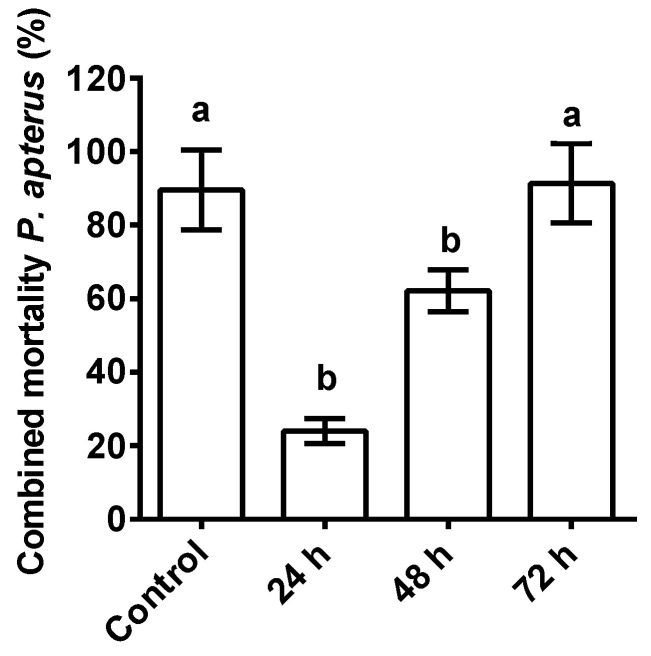
The combined effect of nematode and ASF treatments on host *P. apterus*. Percent mortality of *P. apterus* with combined treatments (ASF-treated nematodes; topical application) at 24, 48, and 72 h. The controls were water-treated (topically applied) and had mean representatives for all time points. Results are expressed as means ± standard deviation (SD). Data represent the means of three replications (*n* = 50). Values (represented as a, b) with different superscripts are significantly different (*p* < 0.05).

**Figure 4 microorganisms-11-01678-f004:**
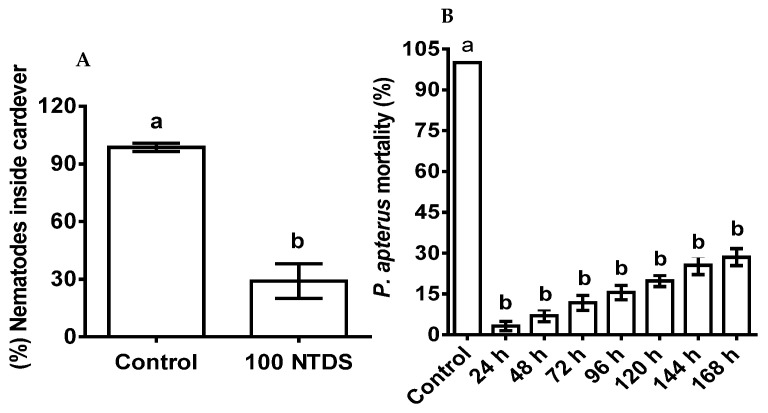
The deterrent effect of nematodes, infectivity, and mortality of *P. apterus* pre-treated with ASF. (**A**) Percent nematode inside a cadaver of 100 mg of pre-treated *P. apterus* after 10 days. (**B**) (%) Mortality of *P. apterus* pre-treated with ASF after 7 days of nematode exposure. Controls (water-treated) are the mean representatives at different time points. Results are expressed as means ± standard deviation (SD). Data represent the means of three replications (*n* = 30). Values (represented as a, b) with different superscripts are significantly different (*p* < 0.05).

**Figure 5 microorganisms-11-01678-f005:**
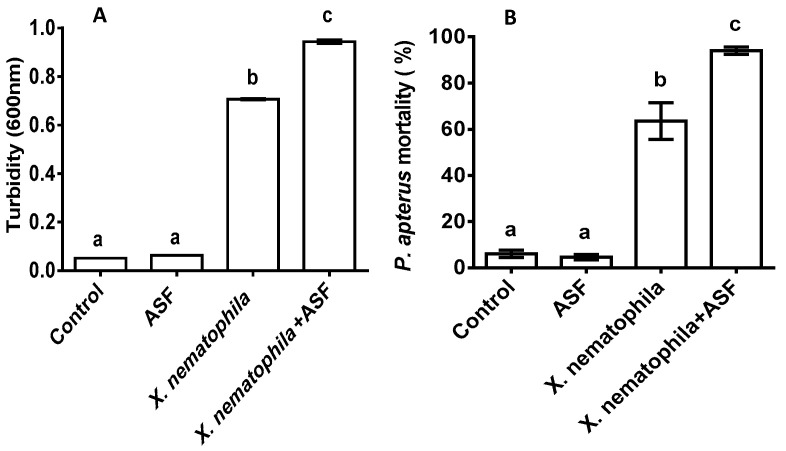
(**A**) In vitro growth of *S. carpocapsae*’s symbiotic bacteria, *Xenorhabdus nematophila*, on ASF-supplemented media and a control with growth media. (**B**) In vitro effect of ASF-treated *X. nematophila* on *P. apterus*’s mortality. Controls were water-treated. Results are expressed as means ± standard deviation (SD). Data represent the means of three replications (*n* = 30). Values (represented as a, b) with different superscripts are significantly different (*p* < 0.05).

**Figure 6 microorganisms-11-01678-f006:**
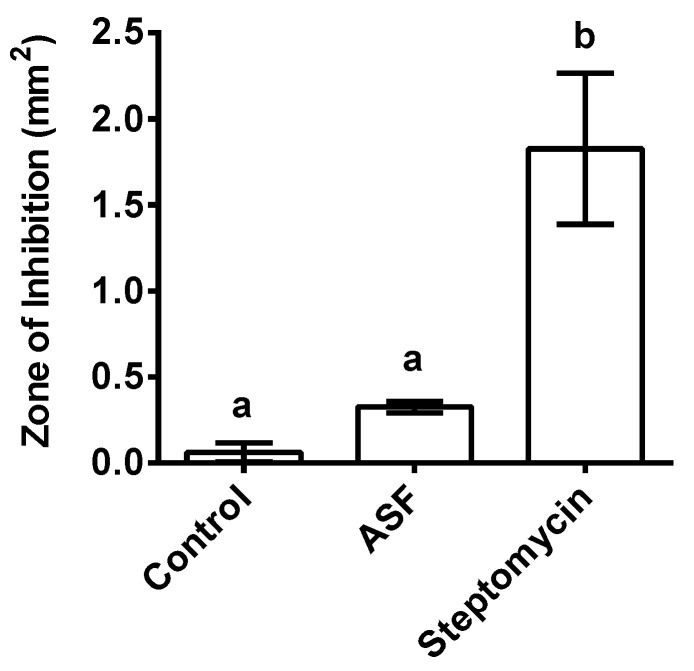
Effect of *X. nematophila* growth on ASF and positive control streptomycin after 3 days of incubation on NA (nutrient agar) plates. Control represents *X. nematophila* growth on NA. The bar graphs are representative of 100 mg of ASF and 30 µg of streptomycin on a paper disc with a size of 0.5 mm placed on NA plates. Each treatment included five replicates (*n* = 5). Points in the bar graphs represent means ± SD. (Dunnett’s test). Values (represented as a, b) with different superscripts are significantly different (*p* < 0.05).

**Figure 7 microorganisms-11-01678-f007:**
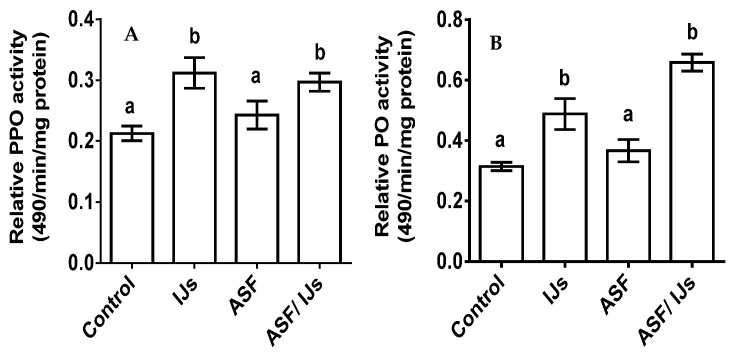
(**A**) ASF effect on the pro-PO activity of *P. apterus*’s hemolymph. (**B**) ASF effect on the PO activity of *P. apterus*’s hemolymph. Control represents water treatment. Results are expressed as means ± standard deviation (SD). Data represent the means of three replications *(n* = 30). Values (represented as a, b) with different superscripts are significantly different (*p* < 0.05).

## Data Availability

The data presented in this study are available on request from the corresponding author. Data are not available publicly due to ethical restrictions.

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
