# Peer review of "Influence of Asafoetida Extract on the Virulence of the Entomopathogenic Nematode Steinernema carpocapsae and Its Symbiotic Bacterium Xenorhabdus nematophila in the Host Pyrrhocoris apterus"

_microorganisms, 2023, doi:10.3390/microorganisms11071678_

Round 1
Reviewer 1 Report
General considerations:
The manuscript (including figure captions) does need a professional linguistic revision by a native English speaker with experience in proof reading manuscripts in entomology or biology. Apart from a few small grammar problems and some typos, the wording is very often unappropriated.
Introduction:
Lines 49-50 ” … The phenomenon happens when EPNs ….”) unappropriated expression.
lines 58-61 Not too related to the objective of the manuscript
Line 75 “ … efficiency in eliminating pest” unappropriated expression
Lines 79-81 review expression
Material and Methods
Is necessary to indicate the number of replicates and repetitions of each assay, this information in some occasions are in the figure captions or in the results section.
Rewrite more of the tittles of the subsections they are not descriptive of its content (ex: “Symbiotic bacterium Xenorhabdus nematophila culture and turbidity test”)
Lines 115-116 “Nematodes were individually transferred to clean water by using a micropipette” is not necessary to explain that you used a micropipette to transfer the nematodes.
Lines 127-128. “The non-wriggling nematodes were considered to be dead”. In the case of S. carpocapsae this is not evidence that the nematodes are death, you must touch the infective juveniles with a needle to check that it does not move and they are dead
Lines 140-141: “… and checked for the number of insects entered into the cadaver” the number of nematodes not insect”. Where are this number of nematodes entered into the cadaver in the results?
Line 143: “P. apterus were treated with ASF 2μl (100 mg)” Is not clear if this treatment was the injection of 2ul ASF into P. apterus.
Results:
Figure y-axis max cannot be 120 or 105 as the maximum percentage is, of course, 100.
The illustrations of the figures 1,2and 3 are irrelevant, deleted.
Figure 3. The mortality of the control at what time is? 24 h? 48h? or 72h? is necessary to indicate this data.
Figure 4 B. The mortality of the control at what time is? 24 h? 48h? …..? is necessary to indicate this data.
In general, is difficult to link the different subsections of results and the subsections of the different assays described in Material and Methods.
Line 255: Where is in M&M the explication of the assay of these results?
Line 281: Not “tropical application” may be topical application.
Where are the results of the “In vivo S. carpocapsae growth” assay?
Rewrite the tittles of the subsections because they are not well related with M&M section (ex: 3.5. Effect of ASF on the In-Vitro growth of S. Carpocapsae symbiotic bacteria Xenorhabdus nematophila and its effect on P. apterus mortality”)
Where is in M&M the description of injecting X. nematophila to P. apterus?
Revised Discussion section there are some sentences that are not in accordance with the results obtained, ex: “In addition to it, both ASF and co-exposed ASF/EPN showed upregulation of the innate immunity markers PPO and PO indicating …” and we can see in Figure 7 that AFS are not different of the control.
References:
Some references are incomplete: ex. 4.- Lines 475-476
Author Response
Comments and Answers:
General considerations:
The manuscript (including figure captions) does need a professional linguistic revision by a native English speaker with experience in proof reading manuscripts in entomology or biology. Apart from a few small grammar problems and some typos, the wording is very often unappropriated.
Response 1: The manuscript has gone through professional English edits and revision by a native English speaker entomologist. We took great affords to correct grammar problems and typo’s throughout the manuscript text.
Introduction:
Lines 49-50 ” … The phenomenon happens when EPNs ….”) unappropriated expression.
Response 2: Accepted and reformed the sentence for appropriate expression.
lines 58-61 Not too related to the objective of the manuscript
Response 3: Thank you. We have deleted the unrelated sentence.
Line 75 “ … efficiency in eliminating pest” unappropriated expression
Response 4: We have reformed and simplified the sentence with appropriate expressions.
Lines 79-81 review expression
Response 5: Thank you. We have now reviewed and clarified the expression.
Material and Methods
Is necessary to indicate the number of replicates and repetitions of each assay, this information in some occasions are in the figure captions or in the results section.
Response 6: Accepted. We have now added this information in the figure captions.
Rewrite more of the tittles of the subsections they are not descriptive of its content (ex: “Symbiotic bacterium Xenorhabdus nematophila culture and turbidity test”)
Response 7: Thank you. We have accepted the suggestion and elaborated the titles of the subsections for more description.
Lines 115-116 “Nematodes were individually transferred to clean water by using a micropipette” is unnecessary to explain that you used a micropipette to transfer the nematodes.
Response 8: Accepted. We have now deleted this information.
Lines 127-128. “The non-wriggling nematodes were considered to be dead”. In the case of S. carpocapsae this is not evidence that the nematodes are death, you must touch the infective juveniles with a needle to check that it does not move and they are dead
Response 9: Yes, it is correct, and we followed the same. We have now modified this information accordingly.
Lines 140-141: “… and checked for the number of insects entered into the cadaver” the number of nematodes not insect”.
Response 10: Thank you. We have corrected it to nematodes.
Where are this number of nematodes entered into the cadaver in the results?
Response 11: It is presented in Fig. 4A.
Line 143: “P. apterus were treated with ASF 2μl (100 mg)” Is not clear if this treatment was the injection of 2ul ASF into P. apterus.
Response 12: Correct, 2ul of ASF prepared from 100mg of ASF with water was injected to P. apterus.
Results:
Figure y-axis max cannot be 120 or 105 as the maximum percentage is, of course, 100.
Response 13: Thanks for the comment. The SD error bar cannot be on the figure if we decide to put only 100%. That the reason for extending the y-axis.
The illustrations of the figures 1,2and 3 are irrelevant, deleted.
Response 14: Accepted. The illustrative figures are now deleted.
Figure 3. The mortality of the control at what time is? 24 h? 48h? or 72h? is necessary to indicate this data.
Response 15: Mortality of water-treated controls are the mean representatives during all the time points. This information is incorporated in the figure legend for clarity.
Figure 4 B. The mortality of the control at what time is? 24 h? 48h? …..? is necessary to indicate this data.
Response 16: Explained above.
In general, is difficult to link the different subsections of results and the subsections of the different assays described in Material and Methods.
Response 17: OK, We have modified and incorporated the details in the present updated manuscript.
Line 255: Where is in M&M the explication of the assay of these results?
Response 18: Please see Section 2.5 (Line 160-162).
Where are the results of the “In vivo S. carpocapsae growth” assay?
Response 19: It is in Section 3.5. “Effect of ASF on symbiotic bacteria X. nematophila growth, turbidity test and its effect on P. apterus mortality”.
Rewrite the tittles of the subsections because they are not well related with M&M section (ex: 3.5. Effect of ASF on the In-Vitro growth of S. Carpocapsae symbiotic bacteria Xenorhabdus nematophila and its effect on P. apterus mortality”)
Response 20: Rewritten.
Where is in M&M the description of injecting X. nematophila to P. apterus?
Response 21: Yes, we have now included the details of X. nematophila injection (Line 159-162).
Revised Discussion section there are some sentences that are not in accordance with the results obtained, ex: “In addition to it, both ASF and co-exposed ASF/EPN showed upregulation of the innate immunity markers PPO and PO indicating …” and we can see in Figure 7 that AFS are not different of the control.
Response 22: Corrected the Typo’s and removed ASF alone. Now the sentence is in accordance with the results obtained. Thanks for the constructive comment.
References:
Some references are incomplete: ex. 4.- Lines 475-476
Response 23: Thank you. We have now corrected it.

Reviewer 2 Report
The Manuscript ID: microorganisms-2399617 examines the impact of Asafoetida extract on the virulence of entomopathogenic nematodes Steinernema carpocapsae and its symbiotic bacterium Xenorhabdus nematophila in the host Pyrrhocoris apterus. It addresses an important and interesting subject especially because many external factors can impact BCAs roles as pesticides, yet the role of Asafoetida extract is unclear. The authors investigated that ASF co-exposed nematodes showed prolonged survival of P. apterus indicating that nematode got attenuated with ASF and recovered over time causing delayed infectivity and mortality. On the other, the growth and proliferation of X. nematophila were not adversely affected by ASF. Moreover, the ASF concentrations used in this study were effective against nematodes rather than X. nematophila. The prophenol oxidase and phenol oxidase upregulation showed that ASF influences the immunity, while EPN/ASF showed a combined immunomodulatory effect in P. apterus.
I’d suggest accepting this paper for publication after further consideration of the following points:
1) The interaction between the EPN/symbiont and any other external factor of interest may be additive, antagonistic, or synergistic. These types of interactions should be stated initially as dependable also on EPN species, insect host, application rate, and time of application.
2) Within this concept, it should be stated that Asafoetida alone is tried as bio-insecticide; e.g., oil extracts from Ferula asafoetida L. as a repellent (Shakeri, 2004)
Shakeri, M. 2004. A review on investigations on pomegranate neck worm in Iran. Proceeding on evaluation of finding and current problems associated with Spectrobates ceratoniae management in pomegranate. Tehran, Iran: Ministry of Jihad-e-Agriculture, Organization of Research and Education. Pp. 20–45.
Also, used against thrips (Noonari et al. 2016):
Noonari, A. M.; Abro, G. H.; Khuhro, R. D. and Buriro, A. S. (2016). Efficacy of biopesticides for management of sucking insect pests of cotton, Gossipium hirsutum (L.). J. Basic Appl. Sci., 12: 306-313.
3) It would be quite perfect if the authors used relevant statistical methods to define the exact type of interaction (additive, antagonistic, or synergistic) in their several studied bio-assays.
4) As EPNs are not evenly distributed in nature or their used microcosms, the authors should refer to this in their procedures. For example, their statement that "Further, 100 IJs were placed on the wet filter paper to establish contact with the experimental and control insects" should be as follows: Further, 100 IJs were placed on the wet filter paper to establish contact with the experimental and control insects considering their spatial distribution (Abd-Elgawad, 2021).
Abd-Elgawad, M.M.M. Optimizing sampling and extraction methods for plant-parasitic and entomopathogenic nematodes. Plants 2021, 10(4), 629. doi:10.3390/plants10040629
5) Some flaws or typo in writings should be corrected; e.g.
i) They show that they can replace synthetic insecticides with equal effectiveness [REF needed], which is widely wanted.
ii) IP should be stated in full at first then abbreviated; i.e., Intraperitoneal (IP) injection
iii) Fig. 1. 1C. Illustration of dead nematodes treated with ASF. It is so dark that EPN are not seen clearly !!
iv) ASF treatment with different dose remained unaffected with P. apterus survival (P < 0.001, F=2.1). Perhaps you mean that “P. apterus survival remained unaffected (P < 0.001, F=2.1) with different doses of ASF treatment”.
v) Then the insects were dissected and checked for the number of insects (do you mean nematodes ?) entered into the cadaver.
Therefore, I would suggest resubmitting after major revision.
Minor editing of English language required
Author Response
Comments and Suggestions for Authors
The Manuscript ID: microorganisms-2399617 examines the impact of Asafoetida extract on the virulence of entomopathogenic nematodes Steinernema carpocapsae and its symbiotic bacterium Xenorhabdus nematophila in the host Pyrrhocoris apterus. It addresses an important and interesting subject especially because many external factors can impact BCAs roles as pesticides, yet the role of Asafoetida extract is unclear. The authors investigated that ASF co-exposed nematodes showed prolonged survival of P. apterus indicating that nematode got attenuated with ASF and recovered over time causing delayed infectivity and mortality. On the other, the growth and proliferation of X. nematophila were not adversely affected by ASF. Moreover, the ASF concentrations used in this study were effective against nematodes rather than X. nematophila. The prophenol oxidase and phenol oxidase upregulation showed that ASF influences the immunity, while EPN/ASF showed a combined immunomodulatory effect in P. apterus.
I’d suggest accepting this paper for publication after further consideration of the following points:
- The interaction between the EPN/symbiont and any other external factor of interest may be additive, antagonistic, or synergistic. These types of interactions should be stated initially as dependable also on EPN species, insect host, application rate, and time of application.
Response 1: Thank you for the concept. We have included this perspective in the
Introduction part. Please see Line: 65-69.
2) Within this concept, it should be stated that Asafoetida alone is tried as bio-insecticide; e.g., oil extracts from Ferula asafoetida L. as a repellent (Shakeri, 2004)
Shakeri, M. 2004. A review on investigations on pomegranate neck worm in Iran. Proceeding on evaluation of finding and current problems associated with Spectrobates ceratoniae management in pomegranate. Tehran, Iran: Ministry of Jihad-e-Agriculture, Organization of Research and Education. Pp. 20–45. Please add in somewhere in the introduction
Response 2: Accepted and added the review in the Introduction section.
Also, used against thrips (Noonari et al. 2016):
Noonari, A. M.; Abro, G. H.; Khuhro, R. D. and Buriro, A. S. (2016). Efficacy of biopesticides for management of sucking insect pests of cotton, Gossipium hirsutum (L.). J. Basic Appl. Sci., 12: 306-313. Please add in somewhere in the introduction.
Response 2: Thank you. Accepted and added the reference.
3) It would be quite perfect if the authors used relevant statistical methods to define the exact type of interaction (additive, antagonistic, or synergistic) in their several studied bio-assays.
Response 4: Since the studies are based on physiological acceptance and rejection of ASF by nematodes and its symbiotic bacteria, we haven’t aimed at the categorised interactions. This perspective is interesting and remains a topic of sequential study. For this manuscript, we considered the basic statistics which provide ample details for the environmental interactions conditions of EPN/Symbiont bacteria/P. apterus.
4) As EPNs are not evenly distributed in nature or their used microcosms, the authors should refer to this in their procedures. For example, their statement that "Further, 100 IJs were placed on the wet filter paper to establish contact with the experimental and control insects" should be as follows: Further, 100 IJs were placed on the wet filter paper to establish contact with the experimental and control insects considering their spatial distribution (Abd-Elgawad, 2021).
: Abd-Elgawad, M.M.M. Optimizing sampling and extraction methods for plant-parasitic and entomopathogenic nematodes. Plants 2021, 10(4), 629. doi:10.3390/plants10040629
Response 5: Thank you. We have referred to this perspective in M&M Section 2.4 (Line 143-144) and added the reference.
5) Some flaws or typo in writings should be corrected; e.g.
- They show that they can replace synthetic insecticides with equal effectiveness [REF needed], which is widely wanted.
Response 6: Accepted. We have now added reference and corrected the sentence.
- ii) IP should be stated in full at first then abbreviated; i.e., Intraperitoneal (IP) injection
Response 7: Thank you. We have now stated in full at first mentioned.
iii) Fig. 1. 1C. Illustration of dead nematodes treated with ASF. It is so dark that EPN are not seen clearly !!
Response 8: Accepted. We have now removed all the illustrative figures for clarity.
- iv) ASF treatment with different dose remained unaffected with apterus survival (P < 0.001, F=2.1). Perhaps you mean that “P. apterus survival remained unaffected (P < 0.001, F=2.1) with different doses of ASF treatment”.
Response 9: Thank you. We have implemented the correction.
- v) Then the insects were dissected and checked for the number of insects (do you mean nematodes ?) entered into the cadaver.
Response 10: We apologise for the Typo. We refer to the number of nematodes entered into the cadaver. The correction is now implemented.
Therefore, I would suggest resubmitting after major revision.
Comments on the Quality of English Language
Minor editing of English language required
Response 11: English language with a native English-speaking entomologist is performed to the revised manuscript.

Round 2
Reviewer 2 Report
Aceepted with minor editing of English language required
Aceepted with minor editing of English language required